# PyOIF: Computational tool for modelling of multi-cell flows in complex geometries

**Iveta Jančigová**[1], **Kristína Kovalčíková**[1], **Rudolf Weeber**[2], **Ivan Cimrák**[1]*

**1** Cell-in-fluid Biomedical Modelling and Computation Group, University of Žilina, Žilina, Slovakia, **2** Institute for Computational Physics, University of Stuttgart, Stuttgart, Germany

* ivan.cimrak@fri.uniza.sk

**Data Availability Statement:** All relevant data are within the manuscript and its Supporting Information files.

**Funding:** KK and IJ were supported by the Slovak Research and Development Agency (contract

## Abstract

A user ready, well documented software package PyOIF contains an implementation of a robust validated computational model for cell flow modelling. The software is capable of simulating processes involving biological cells immersed in a fluid. The examples of such processes are flows in microfluidic channels with numerous applications such as cell sorting, rare cell isolation or flow fractionation. Besides the typical usage of such computational model in the design process of microfluidic devices, PyOIF has been used in the computer-aided discovery involving mechanical properties of cell membranes. With this software, single cell, many cell, as well as dense cell suspensions can be simulated. Many cell simulations include cell-cell interactions and analyse their effect on the cells. PyOIF can be used to test the influence of mechanical properties of the membrane in flows and in membrane-membrane interactions. Dense suspensions may be used to study the effect of cell volume fraction on macroscopic phenomena such as cell-free layer, apparent suspension viscosity or cell degradation. The PyOIF module is based on the official ESPResSo distribution with few modifications and is available under the terms of the GNU General Public Licence. PyOIF is based on Python objects representing the cells and on the C++ computational core for fluid and interaction dynamics. The source code is freely available at GitHub repository, runs natively under Linux and MacOS and can be used in Windows Subsystem for Linux. The communication among PyOIF users and developers is maintained using active mailing lists. This work provides a basic background to the underlying computational models and to the implementation of interactions within this framework. We provide the prospective PyOIF users with a practical example of simulation script with reference to our publicly available User Guide.

This is a *PLOS Computational Biology* Software paper.

## Introduction

### Computational problem description

Microfluidics has been widely adopted by biological and biomedical research fields including (but not limited to) lateral flow tests [1], mixing [2] or cell sorting [3]. Due to the very nature

number APVV-15-0751, www.apvv.sk). IC was
supported by the Ministry of Education, Science,
Research and Sport of the Slovak Republic
(contract number VEGA 1/0643/17, https://www.
minedu.sk/vedecka-grantova-agentura-msvvas-sr-
a-sav-vega/). The funders had no role in study
design, data collection and analysis, decision to
publish, or preparation of the manuscript.

**Competing interests:** The authors have declared
that no competing interests exist.

of the problems, in the design phase, the microfluidic applications may greatly benefit from
simulations. Also, computational models of fluidic systems involving flow of cells and their
manipulation may lead to computer-aided discovery in biology as demonstrated in [4] and the
references therein.

Computational models serve a wide variety of roles, including hypothesis testing, generat-
ing new insights, deepening understanding, suggesting and interpreting experiments, tracing
chains of causation or performing sensitivity analyses. Models cannot replace experiments but
they can demonstrate whether or not a proposed mechanism is sufficient to produce an
observed phenomenon.

To make it easier for people with biological or biomedical background who typically do not
have extensive computational experience to actually use the computational models for these
purposes the usage should be relatively easy. To the best of our knowledge, there is no such
case among the available computer implementations of cell flow models.

With our work we aim to fulfill two goals: 1. We provide a robust and validated computa-
tional model for cell flow modelling. 2. We keep the usage of the implementation easy enough
for non-experts on cell modelling. We make the computer implementation available under
GPL.

## Similar methods and tools

There are numerous computational models that govern flow of cells inside microchannels.
Continuum models treat the membrane as a thin shell and track its deformations in time [5,
6]. The fine network-based models look in detail at the membrane as a lipid bilayer with the
underlying spectrin filament network [7–9]. Coarse-grained models enable simulations of
many cell applications [10, 11]. Mesh-based models study the biomechanical properties of the
cell membrane by means of a triangular network coupled to a lattice-Boltzmann (or any other
fluid solver) representation of the fluid [12–16].

The actual computer implementation of the computational model makes the model use-
ful for the potential user. In-house codes typically have very few users, e.g. in [17], the
authors develop a multiscale and multiphysics computational method to investigate the
transport of magnetic particles as drug carriers in blood flow. The method is implemented
on top of the open-source molecular dynamics simulator LAMMPS [18], however, the code
for the method itself is not available. Some computational methods are published with open
access. An example of such code [19] is a parallel fluid-solid coupling model using the exist-
ing code of LAMMPS for particle propagation of the deformable solids and Palabos [20] for
the simulation of fluid using lattice-Boltzmann method. Other examples include immersed
boundary lattice-Boltzmann finite element method for modelling deformable objects in a
fluid [14, 21] and the work [22] that uses LAMMPS code with GPU accelerated package
using dissipative particle dynamics to implement the model of red blood cell introduced in
[23]. Generally, the usage of computational models is limited only to those experts that
have day-to-day experience with them and these codes have a very long learning curve for
new users. The advantage of PyOIF is the Python interface, which makes the setup relatively
easy.

Several other software tools attempt to provide an easy-to-use interface together with a
well-written documentation. The OpenRBC project [24] aims at modelling of red blood cells
at protein resolution, using multiple millions of mesoscopic particles, thus unsuitable for simu-
lating larger number of cells. Therefore, case studies involving analysis of mesoscopic phenom-
ena, like cell free layer or rheological properties of blood, are impossible to perform using
OpenRBC. Another tool, Hemocell [25], provides implementation of computational model of

red blood cell presented in [26]. The implementation allows simulations of many cells thus, unlike the previous tool, enables analysis of mesoscopic phenomena. The installation of Hemocell requires installation of an external software Palabos.

## Extensible user-friendly tool for computational cell modelling

The PyOIF module fills a niche among the coarse-grained simulation tools. It is open-source, efficiently handles both single and many-cell simulations, requires only simple user input and lightweight scripting in Python and is easily extensible.

# Design and implementation

## Underlying computational model

The PyOIF module is based on two-component model of fluid and immersed objects. The fluid dynamics is governed by the lattice-Boltzmann method (LBM) [27] while the deformable surface is represented by tracking particles propagated with molecular dynamics. Both components are linked via a two-way momentum conserving force interaction. A simplified version of this model was introduced in our earlier work [28].

**Fluid component.**   LBM uses probability distribution functions that propagate and collide over a fixed three-dimensional discrete lattice. We use the existing D3Q19 LBM implementation in ESPResSo that defines 19 discrete directions along the edges and diagonals of the lattice. The particle density functions are defined for each of these directions at every lattice position and time. The propagation is performed using the velocity Verlet scheme and the collision operator satisfies the constraints of mass and momentum conservation.

**Coupling of the model components.**   The fluid is coupled to mesh points by forces. In this coupling, opposite forces exerted on the mesh points and fluid nodes penalize the difference between the local fluid velocity $\vec{u}$ and mesh points velocities $\vec{v}$

$$\vec{F} = \gamma(\vec{u} - \vec{v}). \tag{1}$$

This way, we approximate the no-slip condition.

**Immersed objects component.**   Besides the fluid force, elastic forces are exerted on mesh points that are evaluated from the deformation of the cell. The resultant force $\vec{F}_{tot}$ is the driving force according which the mesh points are propagated in space following Newton's equation:

$$m\frac{\partial^2 \vec{x}(t)}{\partial t^2} = \vec{F}_{tot}, \tag{2}$$

where $m$ is the mass of the mesh points. The sources of $\vec{F}_{tot}$ are the elasto-mechanical properties of the cell membrane, the fluid-structure interaction or possibly other external stimuli.

**Elastic membrane.**   Next we briefly describe the elastic components of the spring network that is formed by mesh points linked together by elastic forces. A more detailed discussion and comparison to similar approaches that use these types of elastic forces, e.g. [16, 29], can be found in [30].

The spring network is a physical model that consists of mesh points located on the surface of the immersed object connected by generalised springs. Part of such spring network is

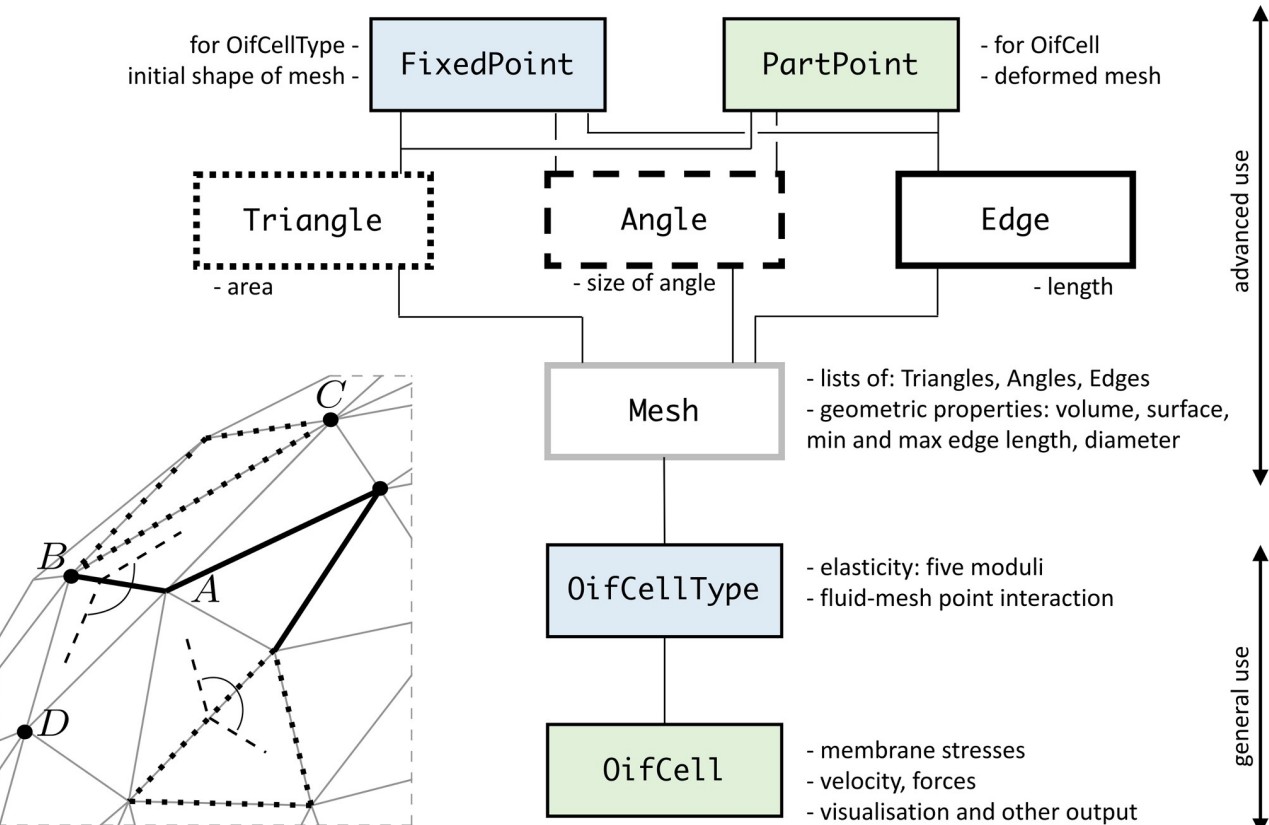

**Fig 1.** Left bottom: Part of the triangular mesh. Geometric entities used to define the elasticity are highlighted: four points (bold dots), three edges (full bold lines), two triangles (dotted lines) and two angles between neighbouring triangles (dashed lines and an arc). Right: Scheme of basic PyOIF classes. Linking of classes and geometrical entities (depicted in the left part of the figure) is emphasized by bold dots (mesh points), full bold lines (edges), dotted lines (triangles), dashed lines (angles) and bold grey lines (mesh).

depicted in the left bottom part of Fig 1. To capture the membrane properties, we use the following moduli acting in the network.

Stretching modulus generates a nonlinear stretching force between two mesh points $A$ and $B$ connected with an edge in the mesh. This force is symmetrically applied at both mesh points and for point $A$ it is defined as

$$\vec{F}_s(A) = k_s \kappa(\lambda) \Delta l_{AB} \vec{p}_{AB}, \tag{3}$$

where $k_s$ is the stretching coefficient, $\vec{p}_{AB}$ is a unit vector pointing from $A$ to $B$, $\kappa$ represents the neo-Hookean nonlinearity $\kappa(\lambda) = (\lambda^{0.5} + \lambda^{-2.5})/(\lambda + \lambda^{-3})$, $\lambda = l_{AB}/l_{AB0}$, $l_{AB0}$ is the relaxed length of the edge $AB$, $l_{AB}$ is the current length, $\Delta l_{AB} = l_{AB} - l_{AB0}$ is the prolongation of this edge.

Bending modulus is derived from the Helfrich energy [31]. It is a potential linked to four mesh points belonging to two neighbouring triangles in a mesh. This energy acts to preserve the angle between the triangles. The expressions for bending force calculations involve position vectors $A$, $B$, $C$ and $D$ of two triangles $ABC$ and $ABD$ that share a common edge $AB$ and

current angle $\theta$, see also left bottom part of Fig 1:

$$
\begin{aligned}
\vec{F}_b(A) &= -k_b \Delta\theta \left( \frac{\vec{N}_C}{|\vec{N}_C|^2} \frac{(A-B, C-B)}{|B-A|} + \frac{\vec{N}_D}{|\vec{N}_D|^2} \frac{(A-B, D-B)}{|B-A|} \right), \\
\vec{F}_b(B) &= -k_b \Delta\theta \left( \frac{\vec{N}_C}{|\vec{N}_C|^2} \frac{(A-B, A-C)}{|B-A|} + \frac{\vec{N}_D}{|\vec{N}_D|^2} \frac{(A-B, A-D)}{|B-A|} \right), \\
\vec{F}_b(C) &= k_b \Delta\theta |B-A| \frac{\vec{N}_C}{|\vec{N}_C|^2}, \\
\vec{F}_b(D) &= k_b \Delta\theta |B-A| \frac{\vec{N}_D}{|\vec{N}_D|^2},
\end{aligned}
\tag{4}
$$

where $k_b$ is the bending coefficient, $\Delta\theta$ is the difference between $\theta$ and $\theta_0$, the angle between these triangles in relaxed state. The vector $\vec{N}_C = (A-C) \times (B-C)$ is the normal vector to triangle $ABC$ and $\vec{N}_D = (B-D) \times (A-D)$ is the normal vector to triangle $ABD$. $(\vec{a}, \vec{b})$ denotes the dot product.

The local area modulus generates forces corresponding to one triangle. The force applied at vertex $A$ of triangle $ABC$ with area $S_{ABC}$ and centroid $T$ is

$$
\vec{F}_{al}(A) = k_{al} \frac{\Delta S_{ABC}}{t_a^2 + t_b^2 + t_c^2} \vec{AT},
\tag{5}
$$

where $k_{al}$ is the local area coefficient, $\Delta S_{ABC}$ is the difference between current $S_{ABC}$ and area $S_{ABC0}$ of the triangle in the relaxed state and $t_a$, $t_b$, $t_c$ are the distances from points $A$, $B$, $C$ to centroid $T$. It has been shown in [32] that this approach is force- and torque-free. Analogous forces are assigned to vertices $B$ and $C$.

The global area modulus ensures that the surface area of the cell remains fairly constant. The application of this force is similar to the local area: we have the proportional distribution according to the distance of vertices from centroid $t_a$, $t_b$, $t_c$. In addition to that we have a weight that takes into account the area of the triangle with respect to the total surface area of the cell:

$$
\vec{F}_{ag}(A) = k_{ag} \Delta S_{cell} \frac{S_{ABC}}{S_{cell0}} \frac{\vec{AT}}{t_a^2 + t_b^2 + t_c^2},
\tag{6}
$$

where $k_{ag}$ is the global area coefficient, $\Delta S_{cell}$ is the difference between the current $S_{cell}$ and area $S_{cell0}$ in relaxed state. The global area contribution described by this formula is due to the triangle $ABC$ with area $S_{ABC}$ and vector $\vec{AT}$. There are other contributions of this kind from other triangles in which $A$ is a vertex.

The volume modulus ensures that the volume of the cell remains fairly constant. Thus, it is also a global modulus, similar to global area. The force as described here, corresponds to triangle $ABC$ and in practice is divided by three and then applied at vertices of the triangle:

$$
\vec{F}_v(ABC) = -k_v \frac{\Delta V_{cell}}{V_{cell0}} S_{ABC} \vec{n}_{ABC},
\tag{7}
$$

where $k_v$ is the volume coefficient, $\Delta V_{cell}$ is the difference between the current volume $V_{cell}$ and volume $V_{cell0}$ in relaxed state. The vector $\vec{n}_{ABC}$ is the unit normal vector to the plane $ABC$, thus the direction of the force is along the triangle's normal pointing inside the cell. The forces are scaled by the triangle area so that the triangle annihilation does not occur.

**Interactions.** In addition to the fluid-object coupling, there are three more types of interactions.

One of them addresses the object-wall encounters. A natural approach in the spring network models is to transform the object-wall interaction into a set of particle-wall interactions. A repulsive force is then applied at the particles when they get too close to the wall. The forces correspond to the soft-sphere potential:

$$V(d) = ad^{-n}, \qquad d < d_{cut}, \tag{8}$$

where $d$ is the distance between the particle and the wall, $d_{cut}$ is the threshold at which this potential starts acting (for larger distances, no force is applied), $a$ is a scaling parameter and $n$ (typically greater than 1) determines how steep the response gets as particles get close to the wall.

The second type of coupling pertains to the object-object interactions, which are transformed into a set of particle-particle interactions. These work similarly to the soft-sphere potential, but take into account not only the distance of the two points but also the normal vectors of the two corresponding objects at these two points. Based on these two vectors, we determine whether the two membranes have crossed each other and apply the membrane collision repulsive forces in the proper direction,

$$V(d) = a \frac{1}{1 + e^{nd}} \qquad d < d_{cut}, \tag{9}$$

where $d$ is the distance between the two particles, $d_{cut}$ is the threshold, at which this potential starts acting, $a$ is a scaling parameter and $n$ determines how steep the response gets as particles approach one another.

Finally, in very confined flows, it is useful to consider also self-cell interactions that ensure that the membrane does not self-overlap. To this end we can again use the particle-based soft-sphere potential.

## Model calibration and validation

The model of cell flow has been validated in terms of comparison to analytical and experimental data.

The calibration of RBC elastic parameters was done using the cell stretching experiment described in [33]. The detailed procedure of calibration and discussion about suitable values of parameters are available in [34].

The fluid-structure interaction in the numerical model is represented by a dissipative coupling parameter. The calibration of this numerical parameter was done in [35].

Red blood cells exhibit rich behavioral patterns in a shear flow. Under certain flow conditions, a red blood cell in shear flow may tumble or exhibit a tank-treading motion of the membrane, depending on the shear rate [36]. We have investigated the inclination angle in [39] and confirmed that our model corresponds to the experimental measurements of the inclination angle reported in [40]. Further, in [34] we performed computational experiments of tank-treading frequency also reproducing the experimental results and thus validating the model.

## Design principles

The model is implemented in the open source software ESPResSo [41] as a module called PyOIF [42]. The computational core of ESPResSo and PyOIF is written in C++ and provides efficient calculations of fluid flow and deformations of elastic objects on both desktop

machines and high performance clusters. The users write simulation scripts in Python, which gives them great flexibility while keeping the code clean and simple.

ESPResSo is a widely used simulation package for research on soft matter, originally developed in 2004 [43] for coarse-grained simulations of charged systems based on particle formulation. Later in 2006 it was extended with lattice-Boltzmann solver for hydrodynamic problems [44] and over the years, functionality such as rigid body mechanics and a lattice-Boltzmann solver running on graphics processors have been added [45]. This setting was ideal for the next step: Grouping particles into objects representing e.g. cells immersed in a fluid, thus creating PyOIF.

The predecessor of PyOIF was OIF framework [46] implemented in Tcl scripting language. OIF was an implementation of a simplified computational model of red blood cells [28]. This model has some modelling issues such as triangle annihilation and non-force-free and non-torque-free calculation of elastic forces. Since its introduction, the model has evolved and these issues have been thoroughly discussed and solved in [30, 32] and the force calculations for three out of five elastic moduli have been changed.

During the year 2018, the ESPResSo package itself underwent a significant transition from Tcl scripting interface to Python. The main changes have been reported in [41]. This was not a merely a translation from one computer language to another, but rather a comprehensive refactoring and restructuring of the whole code base. We followed this transition and we restructured the underlying computational model for cells accordingly. This work resulted in the new Python module PyOIF and involved a completely new object-oriented design that also brought greater memory and computational efficiency.

The best features of the module are its flexibility, extensibility, documentation and simplicity of user input.

*Flexibility* lies in the possibility of setting various computational geometries (including flat or rough walls, periodic obstacle arrays, different cross-sections of microchannels, narrow constrictions). Further, different types of deformable objects (e.g. cells) can be modelled by adapting the elasticity parameters.

*Extensibility* of the module manifests in at several levels. For example, new deformable objects can be introduced by defining their surface triangulation and their elastic or rigid properties. Another example could include more advanced models of cells including their inner structure. Also, the existing classes may be easily extended with further functionality involving analysis of the object's deformation. An advanced user may consider the development of new concepts covering e.g. cell adhesion to surfaces or creation of cell clusters.

*Documentation* provides detailed description of the underlying classes as well as case studies of the module usage. The project webpage features examples of different cell types, different geometries, etc. The book [30] uses the PyOIF module to illustrate various concepts and while it cannot directly serve as a documentation material, it contains detailed explanations and derivation of the underlying model.

*Simplicity* of user input is documented by commented python script available as supplementary material S1 Script.

## General workflow

When performing a simulation, we need to first prepare the surface triangulations of all types of objects that enter the simulation. These can be prepared using various software packages, e.g. Gmsh [47].

In the simulation script we specify the geometry of the domain (size of the simulation box, walls, obstacles), flow conditions (fluid density, viscosity, flow direction, flow velocity), cell

types of all objects (e.g. red blood cells, larger spherical cells with their size, relaxed shape, elasticity), individual cells (position, rotation), their interactions (cell-fluid, cell-wall, cell-cell, self-cell if needed).

Then follows the integration loop, which in addition to the propagation of cells typically includes data output both for visualisation and subsequent analysis.

The simulation script is run by calling the pypresso binary with the script as an argument and may take user defined optional parameters as further input. These parameters are typically the variables of the performed simulation study. During the execution of the python script, the individual commands are parsed and call the corresponding C++ methods. This combination of languages provides user-friendly python interface together with efficient computational C++ core.

Development, preparation and testing of simulation scripts is done on a desktop machine or a laptop, but larger simulation studies are typically performed using HPC clusters. The PyOIF implementation relies on mpi framework for parallelisation. Proper parallelisation can significantly decrease the computational time. For more information and practical usage we refer the user to the documentation [48].

## Classes and architecture

The elastic forces that govern the behavior of the cell membrane imply the basic elements of the computational representation of the cell model, depicted in left bottom part of Fig 1: nodes, edges, triangles and angles. Each of these elements is described by a class, depicted in the right part of Fig 1.

The class `Angle` represents two triangles that share an edge, i.e. it carries information about four nodes. This is useful for two reasons. Firstly, one of the five elastic moduli that we use—the bending force—preserves the angle between two neighboring triangles and thus can use instances of this class to calculate the acting bending forces. And secondly, in order to reduce the number of loops necessary to evaluate the elastic forces, we combine the calculation of all the local forces into one loop over the angles. When looking at a particular angle, besides the bending, the stretching force is calculated for the edge shared by the two triangles and local area forces are calculated for the triangles themselves.

The implementation includes two different types of nodes: `FixedPoint` and `Part-Point`. This allows us to create first an abstract `Mesh` (using instances of `FixedPoint`), which carries information about the object geometry and properties in relaxed state and is represented by `OifCellType`. This information can then be shared by all instances of `OifCell` that have the same properties, but differ in current positions and deformations. Their `Mesh`es are then created as modified copies of the pre-calculated `OifCellType` mesh, but this time with instances of `PartPoint` that have mass and can move in the computational domain.

All interactions happen point-wise. These include cell-wall interactions modeled using the built-in `soft-sphere` potentials at points that get closer to boundaries than a pre-defined threshold.

For cell-cell interactions we developed the `membrane-collision` potential [49]. These interactions calculate the current outer normals for pairs of nodes that belong to two cells that are close to each other. Depending on the relative orientation of the two normals a phenomenological repulsive forces are applied to the nodes. The calibration of this interaction is described in [50].

In confined flows or in other situations when the cells undergo very large deformations, different parts of membrane might get close or in extreme cases even intersect due to the local

nature of the model. To prevent such unphysical behavior, it is possible to include the self-cell interactions as repulsive `soft-sphere` potentials. The `OifCell` class then allows specification of neighbor exclusions so that the potential is not applied to the nearest points on the mesh.

## Input-output

The initial geometry including the shape and size in the relaxed state of one cell (or another object) is specified by two input files: `nodes.dat` and `triangles.dat`. The first file contains triplets of floats (one triplet per line), where each triplet represents the *x*, *y* and *z* coordinates of one node of the surface triangulation. The order of the triplets defines IDs of the mesh points. The second file contains triplets of numbers, this time integers. These refer to the IDs of the nodes in `nodes.dat` file and specify which three nodes form a triangle together. The documentation available at [42] contains further information on how the `OifCellType` ensures that the triangles are properly oriented.

The typical output are user-defined data files (.txt, .dat) that contain information about positions, velocities and other properties of cells that can be post-processed and analyzed.

The second type of output, using the method `cell.output_vtk_pos`, is a .vtk file for visualisation. The PyOIF module allows the user to perform mesh analysis of any elastic object during the simulation. It includes information about acting forces or other local properties, which can also be included in the .vtk output files. Boundaries, walls and fluid can also be saved in .vtk files.

## Units scaling

In ESPResSo, no units are predefined. The user needs to choose a scaling (for example time, length and energy scale) and all other remaining units are derived from these primary choices. For further details on units in ESPResSo we refer to [48], Section Introduction.

Since PyOIF is built upon the core of ESPResSo, we adopt this notion and define the time scale in microseconds, the length scale in microns and the mass scale in $10^{-15}$ *kg*. The units of other physical quantities used in simulations, such as pressure or force, can be derived as proper combinations of the time, length and mass units. In the following, we refer to all these scaled physical quantities as lattice units. Also the numerical values of parameters in the example script are listed in the lattice unit system. We use the SI unit system when referring to the values of parameters measured in biomedical experiments.

With the lattice unit system, the order of magnitude of relevant parameters in our simulations is generally from $10^{-3}$ to $10^3$. However, this is not the only correct way how to set the unit system, and it depends strongly on dimensions and the nature of the simulated phenomenon. Other examples of the unit systems can be found in [30].

## Analysis and visualisation

There are two basic types of output from PyOIF. One serves for visualisation and typically consists of .vtk files that can be visualised using the software ParaView [51]. If the data is in the form of a time series, it can also be exported as a video.

The second type involves quantitative data that is post-processed and analysed. The PyOIF module offers various methods to provide this data. Some of them describe the current geometry of the cell, such as `min_edge_length()`, `max_edge_length()`, `aver_edge_length()`, `surface()`, `volume()`, `diameter()`. Others are related to the cell location and dynamics, such as `get_origin()`, `get_approx_origin()`, `get_velocity()`, `pos_bounds()`. And yet another provides information about local

stresses: `elastic_forces(parameters)`. User can easily add new methods to access cell-related information.

## Geometry considerations

The default configuration of the simulation domain is a box with periodic boundary conditions in all three dimensions. Obstacles are typically simple geometric shapes, e.g. 3-dimensional rhomboids or cylinders. The combination of the periodic boundary conditions of the box and the immersed obstacles allows us to model microfluidic devices with infinite periodic arrays of obstacles. More complex obstacle structures can be modeled by superposition of simpler shapes that may also overlap.

In principle, it is possible to simulate domains that are not periodic, but it requires further thought. In non-periodic domains it might be necessary to re-seed the cells that exit the computational domain and place them at another location (possibly rotated) so that they can enter the domain again as in e.g. [52]. This can be done by the built-in methods for saving the mesh nodes and then using the saved nodes to recreate the cells.

## Extensibility

To demonstrate how a new feature may be implemented in PyOIF, let us assume that we need to identify a location on a cell's surface where the membrane is currently bent the most. Such feature is not included in PyOIF at this time. In geometric terms, this means that the angle between two neighbouring triangles in such spot is an outlier from the relaxed state. In the relaxed state, most angles are close to $\pi$ radians. Therefore, the outliers will be either close to zero or close to $2\pi$ radians. In the former case, the membrane is bent in such a way that it is locally convex, while in the latter case it is locally concave. In what follows, we consider the locally convex case identifying the minimal angle in the mesh. The other case could be implemented in a similar manner.

To demonstrate the simplicity of the needed code, let us implement a new method `min_-angle` for the class `Mesh`. The loop over the angles gives us the minimal angle and the IDs of four points forming the two neighbouring triangles. Then the minimal angle is returned together with the IDs.

```
class Mesh(object):
    def min_angle(self):
      min_angle = 2*numpy.pi
      ids = []
      for angle in self.angles:
        alpha = angle.size()
        if (alpha < min_angle):
          min_angle = alpha
          ids = [angle.A.get_part_id(),angle.B.get_part_id(),
                angle.C.get_part_id(),angle.D.get_part_id()]
      return [min_angle,ids]
```

## Results

To demonstrate the PyOIF capabilities, we present an illustrative case—a simulation of cell flow in bifurcations. It includes a study of cell distribution in the channel. We focus on the explicit details of the simulation script to highlight the connection between the scripting commands and physical setting, as well as on demonstrating how the model can be used to analyse the behavior of many cells in flow.

Besides this case, the PyOIF module has already been used in several application areas. In [53] we have modelled the flow of blood with circulating tumor cells in periodic obstacle arrays. We stated how the density of red cells influences the probability of tumor cell capture. Recently, we have analysed blood cell damage in specific geometries and we developed an indicator for it—the cell damage index [52].

The PyOIF has been further used by external research groups for evaluation of VAD rotatory blood pumps [54], for analysis of micro-roughness and its consequences for platelets activation and platelet catching [55, 56], for simulation of magnetic active polymers for versatile microfluidic devices [57]. Recently, the PyOIF module has been used to investigate the equilibrium structure and quasi-static deformational response of a magnetic polymersome, a hollow object whose magnetoactive part is its shell [58].

## Cell flow in a bifurcation

There are two basic types of bifurcations: a partition of a single flow channel into two daughter branches or a confluence of two flow channels into one.

The diverging bifurcations are responsible for non-uniform partitioning of red blood cells within the network [61]. While there are quantitative data available on the RBC velocity and flux in the daughter branches and it is known that the RBCs have the tendency to enter the daughter branch with the higher flow rate (Zweifach-Fung effect), in some cases an inversion of this effect was observed [62], so further investigation is needed.

The converging bifurcations are helpful in controlling the concentration of cells at a given position across the width of a channel and thus also need to be considered in the process of design of microfluidic devices.

The PyOIF module allows us to include different types of cells in a single simulation. In [67] we modeled a channel that included both a diverging and a converging bifurcation, see Fig 2, and observed the effect of a single larger stiffer cell (e.g. a white blood cell or a circulating tumor cell) on the RBC flow. The commented script `cell_flow_in_bifurcation.py` is provided as part of the supplementary material S1 Script.

**Simulation parameters.**   The simulated domain was a periodic channel with bifurcation into two daughter branches that later converge together. The width of the primary channel was $w_p = 30\mu m$. For the two daughter branches (denoted upper and lower), we considered a symmetric case (C: $w_u = w_l = 15\mu m$) and two asymmetric cases. In one of them the ratio of sizes of daughter branches was 1:2 (A: $w_u = 10\mu m$, $w_l = 20\mu m$) and in the other 2:3 (B: $w_u = 12\mu m$, $w_l = 18\mu m$).

The RBCs in this simulation have a discocyte shape. They are represented by their membranes. The membrane of red blood cell is characterized by its shear modulus, area compression modulus and bending modulus. For the shear modulus we used the value $\mu_0 = 5.5\mu Nm^{-1}$ from the review paper [59]. For the area expansion modulus we worked with $K = 0.025mNm^{-1}$ obtained by dynamic membrane fluctuations [60]. The value of bending modulus was calculated using the biological value $k_c = 1.15 \times 10^{-19}\,Nm$ from [59]. The simulation parameters corresponding to $\mu_0$, $K$ and $k_c$ are `k_s, k_b, k_al`. The correspondence is however not direct. The linking of PyOIF elastic parameters to the mechanical properties of cell membrane is described in detail in [34]. We use values (in lattice units)

$$\texttt{k\_s} = 0.005, \texttt{k\_b} = 0.02, \texttt{k\_al} = 0.007.$$

To ensure that surface and volume of the simulated red blood cell remains fairly constant, we use `k_ag = 0.7, k_v = 0.9` (in lattice units). For the physical quantities we use standard scaling described in Section Design and implementation.

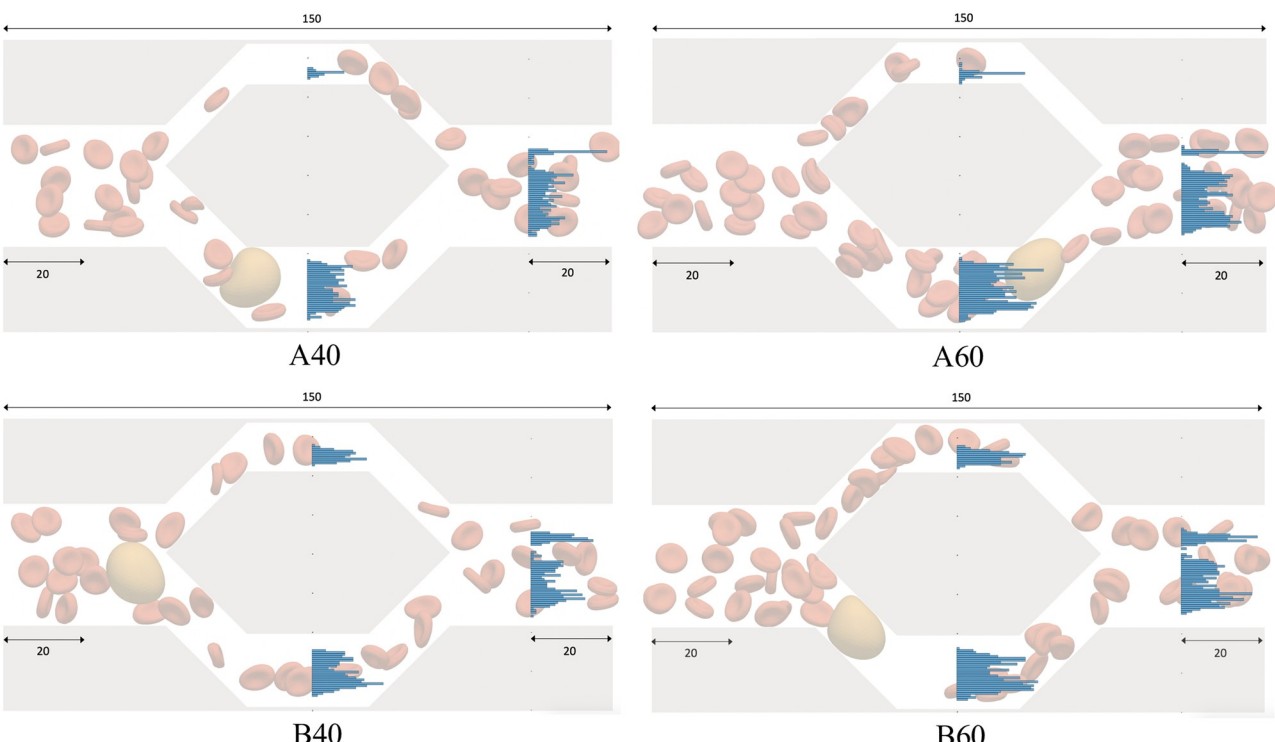

**Fig 2. Distribution of red blood cells in daughter branches and after the confluence.** The histograms indicate $y$-coordinates of RBC centers as the cells crossed the $x = 75\mu m$ and $x = 130\mu m$ positions. The ratio of sizes of daughter branches is 1:2 in case A and 2:3 in case B. In the two cases on the left, there were 40 red blood cells ($Ht \approx 4\%$) and in the two cases on the right, there were 60 red blood cells ($Ht \approx 6\%$). The indicated dimensions are in micrometers. The height of the channel was $h = 20\mu m$.

The definition of CellType is done using the `oif.OifCellType` command which takes several parameters, e.g. the mesh files and the size and elastic parameters of the cell. The cell can be scaled in each of the three dimensions using the parameter `resize`.

To set the correct interaction between the fluid and the particles (forming the spring network) we have to set the frictional coupling parameter $\gamma$. In case we only have one type of cells in the simulation, we can use the `suggest_LBgamma` method of the `oif.OifCell` class that returns the value of $\gamma$ linked to the specific cell based on the analysis from [35]. All types of cells we might want to add to the simulation need to conform to the same $\gamma$ setting. The way to do that is to discretise all new cell types in such a way that the meshes have similar density.

The main integration loop contains outputs for the cell and fluid visualization, integration command containing number of timesteps performed on the C++ level without coming back to the Python level, and the necessary calculations and outputs for the post-processing.

In this simulation, the radius of red blood cells was $3.91\mu m$ and they were represented by meshes with 374 nodes.

In the following we discuss simulations with 40 cells ($Ht \approx 4\%$) and with 60 cells ($Ht \approx 6\%$). Each of these was repeated 10 times with a different random initial seeding of cells. The results are aggregated into six cases by geometry and hematocrit: A40, A60, B40, B60, C40, C60.

The stiffer larger cell was spherical with radius $7.5\mu m$ and in all cases, it was seeded at the center of the parent channel. From the analysis of the dissipative coupling parameter $\gamma$, the discretisation of this cell resulted in 642 mesh nodes.

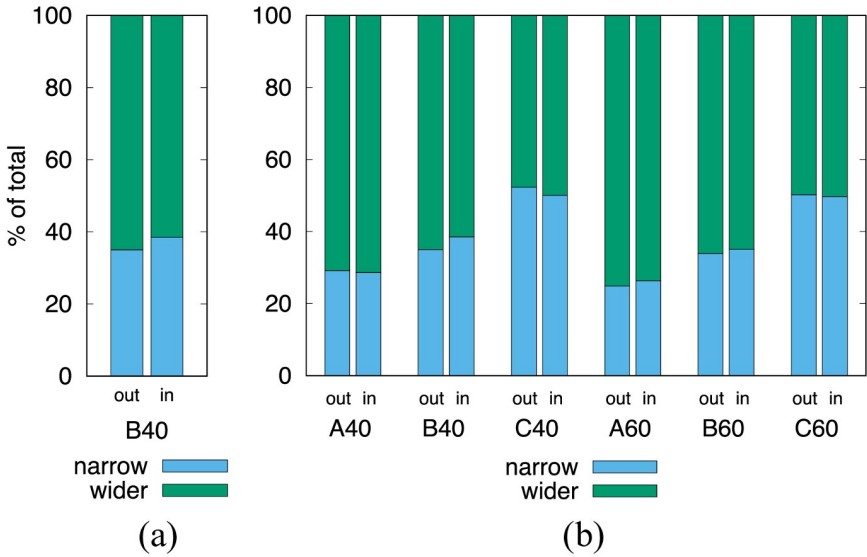

**Fig 3. Percentage split of RBCs as they enter the daughter branches.** (a) The *out* column shows the situation when the rare cell is in the parent channel and the *in* column when the rare cell is in the daughter channel. (b) Comparison of geometries and hematocrits. In the C cases, where both daughter branches have equal width, the one reported as wider is the one the rare cell entered.

The dynamic viscosity of the fluid was $1.5 mPa\,s$, the density $1000 kgm^{-3}$. The stabilized fluid velocity at the center of the parent channel was $1.3 \times 10^{-3}\ ms^{-1}$ similar to the value reported in [65].

**Observations.**   We can observe the distribution of red blood cells in the daughter branches.

As expected, the wider branches receive a larger proportion of red blood cells and we observe a fairly equal split when the daughter branches have the same width, Fig 3. We compare the situations when the rare cell is in the parent branch to the situation when it is inside the daughter branch. In the latter case, for larger hematocrit, the narrow/wider split is more pronounced. For smaller hematocrit values, we see a slight shift both ways, which may indicate that the random seeding influences the split more than the flow itself.

When looking at red blood cells in the asymmetric bifurcations, we observe that they tend to travel closer to the wall towards the wider branch, see Fig 2. The histograms of $y$-coordinates of cell centers shown at $x = 75\mu m$ (middle of the daughter branch) indicate that the cells are the closer to the *right* wall (with respect to the direction of the flow). The double peak distribution in the wider branch is due to the presence of the spherical rare cell. While it travels through the branch, the red blood cells *try to squeeze past it* on the sides.

In addition to local elastic stresses, it is possible to calculate an approximation of fluid force acting on a single red blood cell as a difference between the mesh node velocity and fluid velocity at the mesh node position, summed over all mesh nodes.

Besides quantitative observation, visual inspection of the cell flow may bring additional insights. We present a video of the cell flow as supplementary material S1 Video.

During the simulation, `OifCell` class offers numerous observables such as cell's surface, volume, diameter, different stresses on the surface etc. For a complete list we refer to the PyOIF User Guide. Such data may give valuable insight into the studied problem. To demonstrate this, in Fig 4 we present plots from one particular simulation representing the time

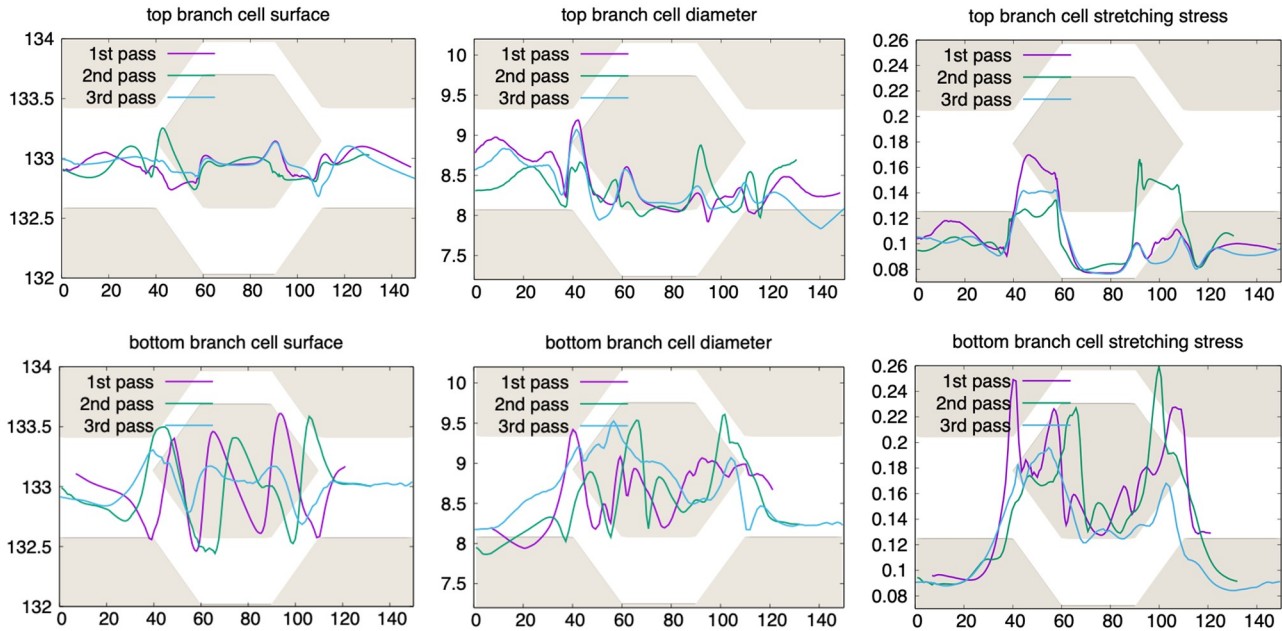

**Fig 4. Quantitative characteristics of cells in bifurcation.** The horizontal axis shows the *x*-coordinate of the cell's center of mass. The cell's surface, diameter and rotation angle during the first pass (purple line), the second pass (green line) and the third pass (blue line) through the bifurcation are shown on the *y*-axis. First row: data for the red blood cell passing through the narrower top branch, second row: data for the red blood cell passing through the wider bottom branch. The bifurcation depicted as a background image helps to link the data to the spatial location of the cell inside the channel.

evolution of the cell's data. From this kind of quantitative output we can for example notice the following interesting observation:

A red blood cell in the narrower branch is less deformed than the one in the wider branch. Indeed, the surface, diameter and volume for the top branch cell oscillates less than for the cell in the bottom branch. This could be counter-intuitive, since the wider branch suggests lower deformation. The explanation lies in the velocity difference: The fluid flows more slowly in top branch than in the bottom branch. This results in lower shear flow in the top branch and thus lower deformation.

We can also observe bulk properties of simulated cells, such as hematocrit shown in Fig 5 and compare different initial conditions. In this case we looked at uniform seeding of 100 red blood cells in which the elastic properties of all cells corresponded to those calibrated by the stretching experiment and at mixed seeding, in which half of the cells were significantly stiffer. The stiffer cells may represent those affected by disease, e.g. malaria.

In both cases, the cells were seeded at the same positions, but the seeding was not spatially-homogeneous, i.e. there was a cell-free spherical area in the main channel that gradually dissipated. In the uniform seeding chart we can observe the complementary oscillations of hematocrit values in the parent channel and the wider lower branch, as the cell-free area moves with the flow until it dissipates.

Interestingly, with the mixed seeding the hematocrit values stabilize sooner than with the uniform seeding. Observations such as this one can serve as a starting point for further investigation of how blood properties influence flow in various geometries.

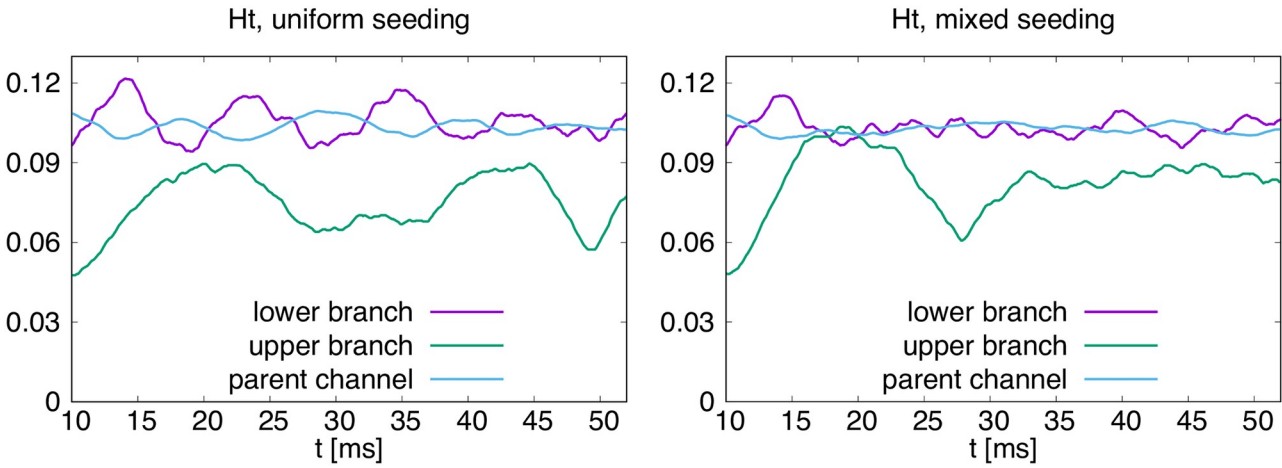

**Fig 5. Hematocrit in the main parent channel and two daughter branches averaged over 10ms.** In the uniform seeding, all red blood cells had the same elastic properties as calibrated using the stretching experiment. In the mixed seeding, the cells were placed at the same starting positions, but randomly selected half of them were significantly stiffer.

**Time complexity.** In this section we present an overview of the CPU time for this simulation. All the simulations were run on CPUs of a Threadripper 2950X machine with 16 cores and 32 logical processors with 32 GB RAM and 1x Samsung 970 Evo NVME SSD.

To show the scalability of the computations, we performed test simulations with various numbers of cells on various numbers of computational cores. We used 0, 5, 10, 20, 40 and 60 cells and we parallelized the computation onto 1, 3, 5, 6, and 10 cores by splitting the simulation box in $x$ or in both $x$ and $y$ directions. Table 1 shows the CPU time in seconds for simulating 0.1 $ms$ of the cell flow averaged over 2$ms$. The data in the parentheses show the CPU time devoted to one blood cell. It was computed by subtracting the CPU time for fluid only from the respective CPU time for $n$ cells and fluid, divided by $n$. In this simulation, the fluid was discretized with spatial grid of 1$\mu m$ and each RBC had 374 mesh nodes.

From the table we can clearly see that increasing the number of processors decreases the CPU time and increasing the number of cells increases the CPU time. We calculated the parallel efficiency $E$ and in Fig 6 we depict the values $E = S/N$, where $S$ is the speed-up ratio $S = T_1/T_N$, $N$ is the number of computational cores and $T_N$ is the CPU time for simulation with $N$ computational cores. We can see that parallel efficiency drops with the increasing number of cores to around 60% with 10 cores.

To provide the reference time consumption, we measured the time needed to simulate one pass of the rare cell through the bifurcation channel which amounts to approximately 138$ms$

**Table 1. CPU times in seconds for different number of cores and number of cells for 1000 simulation time steps which correspond to 0.1$ms$ of cell flow.** The corresponding time per one blood cell is given in parentheses. The reference simulation is in bold.

| | | | n_cells | | | |
|---|---|---|---|---|---|---|
| n_proc | 0 | 5 | 10 | 20 | 40 | 60 |
| 1 | 48.14 | 55.90 (1.55) | 62.68 (1.45) | 84.37 (1.81) | 145.00 (2.42) | 242.47 (3.23) |
| 3 | 17.40 | 21.27 (0.77) | 22.70 (0.53) | 32.14 (0.74) | 53.56 (0.90) | 93.56 (1.26) |
| 5 | 12.79 | 16.88 (0.81) | 19.13 (0.63) | 23.78 (0.55) | 37.35 (0.61) | 58.97 (0.76) |
| 6 | 11.02 | 14.82 (0.76) | 15.98 (0.49) | 22.90 (0.0.59) | **36.04 (0.62)** | 53.31 (0.70) |
| 10 | 8.31 | 11.61 (0.66) | 13.96 (0.56) | 18.31 (0.50) | 26.67 (0.45) | 40.22 (0.53) |

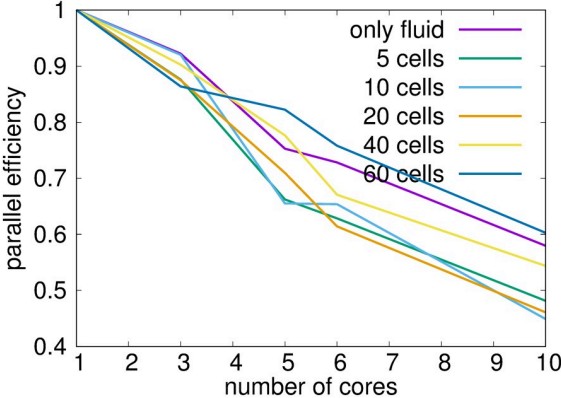

**Fig 6. Parallel efficiency of the code.**

of flow. With the *x*-distance $150\mu m$ traveled by the rare cell, this leads to the average velocity of $1.085mm/s$. The reference simulation was parallelized onto 6 cores and took 16 hours.

## Availability and future directions

Since the PyOIF module is part of the ESPResSo package, it is available from the github repository [71] forked from the main ESPResSo branch [72] and it is distributed under the terms of the GNU General Public Licence. The ESPResSo code is developed and maintained primarily at the Institute for Computational Physics of the University of Stuttgart, Germany, but has contributors from all over the world. The PyOIF is developed and maintained at Cell-in-fluid Biomedical Modelling and Computations Group [73] at University of Zilina, Slovakia. We are open to cooperation with other contributors who would like to add new features to the PyOIF module. Their work would be integrated into the main package using the same processes as the current developers use, i.e. pull requests to review new features and bug fixes.

The community actively uses two mailinglists: espressomd-devel@nongnu.org for developers and espressomd-users@nongnu.org for users. Both work on a subscription basis, are archived and are a good starting point and resource for anybody who wants to actively use or develop this computational tool. The questions related to PyOIF module are typically answered by its developers who are actively using it. There has also been development of new features per users' requests.

The source code available at the repository contains sample examples with simple simulations of cells. More advanced examples involving cell random seeding, testing of cell-wall interactions, stretching tests of red blood cells are available at PyOIF webpage [42]. We strongly recommend following the steps in the user guide to install PyOIF. And as any scientific software, this module should not be used as a black box. The users should be familiar with the underlying model and meaning of the parameters they use.

Currently, there are two features under development:

- Inclusion of the cell nucleus: While not applicable to red blood cells which do not posses one, this will allow us to model other type of cells. Two different approaches are being evaluated, one using bonded and one using non-bonded potentials [74].

- Variable viscosity: Currently, both the fluid and the inside of immersed objects have the same (constant) viscosity. In blood, this is not the case. The blood plasma has about five

times lower viscosity than the fluid inside the red blood cells. We are implementing an option that would allow setting a different viscosity inside the immersed objects [75].

The near-future plans include the development of a checkpointing system that would allow saving the the complete state of an elastic object. Currently, we can save and reconstruct the geometry, but not the acting forces, which introduces certain inaccuracy when re-seeding cells.

To conclude, PyOIF is a user-friendly powerful computational tool that has already been successfully used for simulations of blood flow in microfluidic devices and can be used for predictive simulations of elastic objects in flow. We hope that it will become a valuable resource both for computational and experimental biomedical community.

## Supporting information

**S1 Script. Commented simulation script, including input files.**
(ZIP)

**S1 Video. Video of cell flow through bifurcation.**
(MP4)

## Acknowledgments

We thank Martin Slavík for his contribution to the rewriting of the Tcl code into Python and all the members of the Cif-BMCG research group for extensive testing of PyOIF.

## Author Contributions

**Conceptualization:** Iveta Jančigová, Rudolf Weeber, Ivan Cimrák.

**Formal analysis:** Rudolf Weeber, Ivan Cimrák.

**Funding acquisition:** Ivan Cimrák.

**Investigation:** Iveta Jančigová, Kristína Kovalčíková, Ivan Cimrák.

**Methodology:** Iveta Jančigová, Kristína Kovalčíková, Ivan Cimrák.

**Software:** Iveta Jančigová, Ivan Cimrák.

**Supervision:** Ivan Cimrák.

**Validation:** Iveta Jančigová, Kristína Kovalčíková, Ivan Cimrák.

**Visualization:** Iveta Jančigová, Ivan Cimrák.

**Writing – original draft:** Iveta Jančigová, Kristína Kovalčíková, Ivan Cimrák.

**Writing – review & editing:** Iveta Jančigová, Kristína Kovalčíková, Rudolf Weeber, Ivan Cimrák.

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
