## [Decision Letter · Decision Letter 0]

12 May 2020

Dear Prof. Cimrak,

Thank you very much for submitting your manuscript "PyOIF: Computational Tool for Modelling of Multi-Cell Flows in Complex Geometries" for consideration at PLOS Computational Biology.

As with all papers reviewed by the journal, your manuscript was reviewed by members of the editorial board and by several independent reviewers. In light of the reviews (below this email), we would like to invite the resubmission of a significantly-revised version that takes into account the reviewers' comments.

I suggest you shorten the paper, so it doesn't include in-depth presentations of previously published methods and results. Instead, please focus on the software, usage, and capabilities to provide the user information that goes beyond published methods and tutorials.

We cannot make any decision about publication until we have seen the revised manuscript and your response to the reviewers' comments. Your revised manuscript is also likely to be sent to reviewers for further evaluation.

Sincerely,

Dina Schneidman-Duhovny

Software Editor

PLOS Computational Biology

Reviewer's Responses to Questions

**Comments to the Authors:**

Reviewer #1: The paper presents a open source software for the simulation of red blood cells in flows. The model is based on a Lattice Boltzmann approach for the fluid part and a spring-based approach couple to the LB for the modelling of the soft matter. The paper starts with an overview of the model, followed by its validation and its application to 3 different cases. While the interest and availability of such a model is surely important for the biology community, the paper itself, is, in my opinion, simply a compilation of the previous authors' works. For example, the model calibration and validation is purely based on previous published papers and thus the section could be largely shorten. The general workflow section is based on the user guide documentation already available online with the source codes, while the design principles sound more a compilation of the advantages of the code provided with technical justification and/or evidence. In section 3, results, line 398, page 11, the authors explicitly state that those results 'have been thoroughly described elsewhere'. While they do not add the corresponding references, my question is thus as followed: what is the benefit for the community to have a paper repeating results that have already been published? For example, the case of section 3.1 has been published in ref[34]. Section 3.3 states that the authors 'are able to track more detailed information' (line 605, page 17), however, none of the listed information is actually presented in the paper. Moreover, only a video is provided and no quantitative data presented nor any validation for this case. Also, from the video, it looks like the cells are spherical and do not have a discocyte shape for the most common shape of a healthy RBC. Can the authors explain if the code handles discocyte and other RBC shapes?

While the proposed tool has some significance for the community, I do not think the paper is acceptable for publication in PLoS Computational Biology as I can't clearly see any new understanding of the physics of RBC motion and the model itself has already been published.

Reviewer #2: The manuscript introduces a software package for simulations of deformable cells in flow. Fluid flow is modeled using lattice Boltzmann method, whose implementation is performed within the ESPRESSO package. Deformable particles or cells are modeled using discrete network-based model, which is implemented in the new PyOIF package. The models are based on state of the art developments, and their implementation should be useful for researchers working on modeling deformable particles and cells in fluid flow. The main advantage of the described software package is its open access, as only a few open packages of potentially similar capabilities are available at the moment. It is difficult to estimate whether this software package will generate a lot of interest, but I think it deserves to be announced (or published). From the description, it sounds that the authors are planning its continuous maintenance and improvement, and they also provide some help and information to the users. I am in favor of publishing this manuscript, and would like to ask the authors to address a few rather minor concerns.

1) The authors mention a couple of software packages, which may have similar/comparable possibilities. I think it would be useful to discuss similarities/differences advantages/disadvantages in more detail. Why should I choose the proposed software over another one? Maybe some sort of a summarizing table would be useful.

2) Clearly, sophisticated models are complex and have a number of parameters. Therefore, it is probably impossible to use them as black boxes and some level of understanding is needed. The authors mention several times that certain parameters were calibrated in previous works. Does a user need to understand all these previously performed tests, in order to have a good idea of possible pitfalls? Would the software give some suggestions or warnings?

3) It is difficult to judge software performance. Therefore, I would like to ask the authors to mention explicitly the performance numbers for different tests. How long each simulation takes? How well it is paralleled (speed-up, strong and weak scaling)? What is the most expensive part LBM or cell modeling? I think it would be useful to give running times for different examples to have an idea of the performance.

4) Are you planning collective software development, so that users can contribute to this? How can you enable this efficiently?

**Have all data underlying the figures and results presented in the manuscript been provided?**

Reviewer #1: Yes

Reviewer #2: Yes

PLOS authors have the option to publish the peer review history of their article (what does this mean?). If published, this will include your full peer review and any attached files.

Reviewer #1: No

Reviewer #2: Yes: Dmitry Fedosov
---

## [Decision Letter · Decision Letter 1]

14 Aug 2020

Dear Prof. Cimrak,

We are pleased to inform you that your manuscript 'PyOIF: Computational tool for modelling of multi-cell flows in complex geometries' has been provisionally accepted for publication in PLOS Computational Biology.

Best regards,

Dina Schneidman

Software Editor

PLOS Computational Biology

Reviewer's Responses to Questions

**Comments to the Authors:**

Reviewer #2: The authors have given satisfactory answers to all of my concerns, so I have no further comments and recommend the manuscript for publication.

**Have all data underlying the figures and results presented in the manuscript been provided?**

Reviewer #2: Yes

PLOS authors have the option to publish the peer review history of their article (what does this mean?). If published, this will include your full peer review and any attached files.

Reviewer #2: No

---

## [Editor Report · Acceptance letter]

8 Oct 2020

PCOMPBIOL-D-20-00375R1 

PyOIF: Computational tool for modelling of multi-cell flows in complex geometries

Dear Dr Cimrak,

I am pleased to inform you that your manuscript has been formally accepted for publication in PLOS Computational Biology. Your manuscript is now with our production department and you will be notified of the publication date in due course.

With kind regards,

Laura Mallard
